# Ultraviolet-A Radiation (UV_A_) as a Stress and the Influence of Provenance and Leaf Age on the Expression of Phenolic Compounds by *Eucalyptus camaldulensis* ssp. *camaldulensis*

**DOI:** 10.3390/plants14030493

**Published:** 2025-02-06

**Authors:** Santosh Khanal, Simone J. Rochfort, Martin J. Steinbauer

**Affiliations:** 1Department of Ecology, Environment and Evolution, School of Life Sciences, La Trobe University, Bundoora, Melbourne, VIC 3086, Australia; 2Agriculture Victoria, AgriBioscience, Bundoora, Melbourne, VIC 3083, Australia; s.rochfort@latrobe.edu.au; 3School of Applied Systems Biology, La Trobe University, Bundoora, Melbourne, VIC 3086, Australia

**Keywords:** ecophysiology, environmental gradients, leaf ontogeny, leaf pigments, leaf tagging, photodamage, photoprotection, river red gum, tannins

## Abstract

Ultraviolet radiation (UV) represents a significant abiotic stress, affecting green plants. Phenolic compounds have been suggested as components involved in plant photoprotective adaptation. We used a unique combination of experimental (LED lighting and leaf tagging) and analytical (unbiased, or untargeted, metabolomics) approaches to study the effects of high (approximating mid-summer) and low (approximating winter) levels of UV_A_ on the expression of phenolic compounds. These consisted of river red gum (*Eucalyptus camaldulensis* ssp. *camaldulensis*) of five provenances. The geographically separated provenances used in our study spanned the lowest and highest latitudes of the range of this subspecies. The concentrations of gallotannins and ellagitannins (i.e., hydrolysable tannins) increased most under high levels of UV_A_, but responses only differed slightly among provenances. The most substantial changes in the composition of phenolic compounds were associated with leaf age. Overall, 3-month-old (herein, termed ‘young’) leaves had substantially different phenolic compositions to 6- and 12-month-old (‘old’) leaves. Hydrolysable tannins were more abundant in young leaves, whereas pedunculagin, catechin, and kaempferol galloyl glucoses were more abundant in old leaves. High levels of UV_A_ altered the expression of phenolic compounds, but our experimental saplings were unlikely to experience photoinhibition because they were not exposed to high levels of light and low temperatures, nor were they nitrogen-limited. We expect that changes in phenolic compounds would have been more pronounced if we had induced photoinhibition.

## 1. Introduction

Plants can endure diverse abiotic and biotic stressors by altering their physiology and biochemistry, which may also change their metabolome. Plant secondary metabolites (PSMs), including phenolic compounds, can exhibit qualitative and/or quantitative changes in response to regulatory processes and thereby mitigate the impacts of potentially harmful stressors [1,2,3,4,5,6]. The identification of metabolites, mediating the plant tolerance of stress, is essential to understanding adaptive responses. Unbiased (or non-targeted) metabolomics has emerged as a powerful tool with which to investigate plant responses because it expedites the identification of metabolites of potential adaptive significance [7,8].

Phenolic compounds are the most abundant and widely occurring PSMs of the plant kingdom [9,10]. “Tannins” are possibly the most studied of the phenolic compounds. This is because they are implicated (rightly and wrongly) in many interactions between plants and herbivores [11,12,13,14]. Structurally, phenolic PSMs comprise at least one aromatic ring, with one or more hydroxyl group. The interactions between these two structural moieties are key to biological activity as they relate to the ability to produce free radicals, where the radical is stabilised by delocalisation [9,10,15]. High-molecular-weight tannins (e.g., the hydrolysable and condensed tannins) are up to 15–20 times more effective in quenching peroxyl radicals than simple polyphenols [16]. Oxidative stress arising from photodamage has been suggested to be at least as important, if not more so, to the evolution and proliferation of plant phenolic compounds as herbivory [17]. The importance of phenolic compounds for the photoprotection of eucalypt leaves was first proposed by Close and McArthur [17].

UV radiation is a known elicitor of the synthesis of PSMs [18]. Almost 95% of UV_A_ and 5% of UV_B_ radiation penetrates the Earth’s atmosphere and reaches the surface. Although UV_A_ is less damaging than UV_B_, it can penetrate deeper and is more abundant. Hence, UV_A_ can cause significant damage to plant tissues and DNA [19,20]. The lower cloud cover over Oceania and Antarctica compared to the rest of the world means that there is a significantly greater flux of UV_A_ in these regions compared to elsewhere on the planet [21]. Therefore, plants at these latitudes are exposed to higher UV_A,_ with UV flux increasing by approximately 1–2% per one degree of latitude away from the equator [22,23]. Although plants are exposed 10–100 times more UV_A_ than UV_B_, most research has focussed on the effects of UV_B_.

Phenolic compounds have unique absorption maxima and can absorb UV radiation without affecting the availability of photosynthetically active radiation (PAR). For example, hydroxycinnamic acid exhibits maximum absorption values between 310 and 332 nm, maximum flavone levels between 250 and 270 and 330 and 350 nm, and maximum flavonol levels between 250 and 270 nm and 350 and 390 nm [24]. Studies on the effects of increased UV on a variety of non-woody, herbaceous plants, and fruits are numerous because phenolic compounds can affect the colour and/or flavour of edible tissues to humans [25]. Such studies often report increased concentrations of a range of phenolic compounds [26,27,28]; however, some have reported no change in composition [29].

For such an ecologically and economically important group of trees, the paucity of information about the responses of eucalypts (an informal grouping encompassing the genera *Eucalyptus*, *Angophora*, and *Corymbia* and others) to abiotic stressors is surprising, in particular their tolerance to the harsh Australian sun. Since our lab was already researching natural variation in foliar phenolics of river red gum, *Eucalyptus camaldulensis* [30,31], we chose to conduct an experimental study to quantify how this widely distributed species responds to a specific component of solar radiation, namely UV_A_, at low and elevated fluxes. Rather than use conventional analytical techniques to assess changes in phenolic composition, we chose to employ an unbiased metabolomics approach. Such an approach had not been applied to eucalypt phytochemistry before.

## 2. Results

The leaves of *E. globulus* and, to a lesser extent, *E. nitens* have been reported to develop epidermal blisters and exhibit curling when grown under elevated humidity levels, as can occur in glasshouses [32]. We did not observe any similar morphological responses by the leaves of our saplings, which could suggest they experienced ambient humidities comparable to those experienced under non-glasshouse conditions. Alternatively, *E. camaldulensis* might not respond similarly to the two aforementioned species to high humidity.

### 2.1. Effect of Provenance on Phenolic Composition

A principal component analysis (PCA) ordination of the phenolics of the individual saplings used in the experiment revealed some differentiation according to provenance (Figure 1). The variation explained by the first principal component (PC1) was only 3.8%, while the variation explained by the second principal component (PC2) was also small, at only 3.0%. PC1 provided the greatest separation of G_1_ (latitude −31.46 S) and G_2_ (latitude −35.75 S), and PC2 provided the greatest separation of G_2_ and G_5_ (latitude −33.10 S).

*K*-group analysis identified five phenolic metabolites that were differentially expressed among the provenances and which had high loadings. Some of these metabolites were tentatively identified as pentagalloylglucose (m/z 939.1107), galloylglucose (m/z 331.0669), a quercetin derivative (m/z 433.0408), HHDP-galloylglucose (m/z 633.0735), and an unknown phenolic compound (m/z 337.2052). Pentagalloylglucose occurred at the highest abundance in G_1_ and occurred at the lowest abundance in G_2_. The quercetin derivative occurred at the highest abundance in G_1_ and occurred at the lowest abundance in G_4_ (latitude −33.85 S).

### 2.2. Effect of UV_A_ on Phenolic Composition

PCA ordination showed the clear separation of saplings according to the light treatments they were grown under (Figure 2A). For PC1 (9.3% variation), samples from high- and low-light treatments clustered apart from the samples obtained from the control and ambient light treatments. Re-modelling the data using only the samples from the high and low light treatments showed the clear separation of phenolic metabolomes (Figure 2B). However, the PC1 of this second ordination only explained 5.9% of the variation in the data.

*K*-group analysis identified 20 compounds based on their effect size (Table 1). Additional MS/MS tandem mass spectrometry and fragmentation enabled 17 of these phenolic compounds to be tentatively identified (Table 1). Tukey’s multiple comparisons of means revealed that 12 hydrolysable tannins (comprising ellagitannins and gallotannins) occurred in significantly higher concentrations in leaves grown under the high-UV_A_ treatment (Figure 3).

### 2.3. Effect of Leaf Age on Phenolic Composition

Leaf age was the most influential factor in the differential expression of phenolic compounds (Figure 4). The phenolic metabolomes of 6- and 12-month-old leaves (PC2 1.9% variation) were more alike than those of 3-month-old leaves (PC1 23.2% variation). Since the separation between 6- and 12-month-old leaves was comparatively small, we combined these data. Hereafter, we refer to them ‘old leaves’. We refer to the three-month-old leaves as ‘young leaves’. As determined based on fold change and *p*-values, 14 phenolic compounds differed significantly in concentration between the two age classes (Table 2). Concentrations of pedunculagin, catechin, and kaempferol galloylglucoses were higher in old leaves, whereas concentrations of hydrolysable tannins and quercetin were higher in young leaves (Figure 5).

## 3. Discussion

Through our unique experimental and analytical approaches, we obtained important new insights into the effect of light intensity and quality on the expression of phenolic compounds by Australia’s most iconic and globally significant species of tree. The only other study to investigate the effect of UV_A_ on the composition of eucalypt foliar phenolics is that by Close et al. [33]. Nevertheless, these authors [33] only excluded UV_A_ radiation, rather than replicating different intensities of exposure. Moreover, these authors studied a species of eucalypt (namely shining gum, *Eucalyptus nitens*) with a restricted region of endemism in high-altitude, frost-prone habitats in the Australian Alps and which might therefore be expected to be better adapted to higher levels of UV_A_ than *E. camaldulensis*.

### 3.1. Effect of Provenance on Phenolic Composition

We found that foliar concentrations of pentagalloylglucose and quercetin (a flavonol) were higher in plants with provenances from lower latitudes, e.g., G_1_ from Wilcannia (average annual rainfall 264 mm) and G_3_ from Boolcunda Creek (average annual rainfall 306 mm). These provenances of *E. camaldulensis*, in particular, evolved under conditions of very low average annual rainfall and pronounced drought. Pentagalloylglucose is reported to be an effective 1,1-diphenyl-2-picrylhydrazine (DPPH) radical scavenger and to have better free radical scavenging properties than vitamin C [34]. Plant flavonols, such as quercetin, flavonoids, and anthocyanins, have strong radical scavenging activities and, when in elevated concentrations, can contribute to the mitigation of drought stress [3,35,36]. In the morphologically juvenile leaves of a selection of provenances of *E. globulus* and *E. viminalis*, a mild water deficit induced an increase in the concentrations of condensed tannins [5]. These authors also reported that trait plasticity in response to water deficit was less pronounced than that of constitutive trait variation among provenances. We suggest that a similar explanation could underpin the relatively small influence of provenance we found in our study.

### 3.2. Effect of UV_A_ on Phenolic Composition

Our results revealed an increase in the concentrations of phenolic acids, hydrolysable tannins, and flavonoids under high levels of UV_A_. Close et al. [37] reported two periods of elevated concentrations of total anthocyanins in the leaves of nutrient-deprived, seedling *E. nitens*. These corresponded to photoinhibitory growing conditions, i.e., low ambient temperatures and high levels of light. The concentrations of galloylglucose and flavonols were not related to the anthocyanin concentration at these times, which the authors suggested negated the possible role of anthocyanins in foliar antioxidant activity. Rather, Close et al. [37] proposed that the presence of higher concentrations of anthocyanins in nutrient-deprived seedlings during photoinhibitory conditions supported their role in light attenuation. In contrast, Tian et al. [38] suggested that pentagalloylglucose provided photoprotection to the leaves of *Eucalyptus* [*globulus*] ssp. *maidenii*. At least three studies reported enhanced biosynthesis of quercetin under UV_B_ stress conditions, suggesting their active role in photoprotection [39,40,41]. Chloroplast-located quercetin and its glycosides are known for their role as effective singlet oxygen quenchers generated by excess blue light [42]. 

### 3.3. Effect of Leaf Age on Phenolic Composition

Perhaps unsurprisingly, we found that leaf age had the greatest influence on the composition of phenolic compounds in our study species. The contrast in phenolic composition was most pronounced when leaves were 3 months old compared to when they were either 6 or 12 months old. Overall, 3-month-old leaves had higher concentrations of phenolic acids, flavonoids, quercetin, and hydrolysable tannins (both galloylglucose and ellagitannins), whereas 6- and 12-month-old leaves had higher concentrations of glycosylated hydroquinone, kaempferol galloylhexoside, and pedunculagin. The photosynthetic apparatus of leaves is not fully functional when they are young. Hence, it requires protection [43]. In some eucalypts, including *E. camaldulensis*, anthocyanins attenuate light intensity in young leaves before the concentration of chlorophylls rises, thereby increasing their photosynthetic capacity [44]. This can give the young leaves of some species of eucalypt obvious red colouration. Since anthocyanins and phenolic compounds share the same biosynthetic pathway (the phenylpropanoid pathway), young red leaves are likely to produce non-pigment compounds with photoprotective properties, e.g., tannins [37,45]. McArthur et al. [46] reported that the total phenolic concentration of *E. nitens* was not attained until seedlings were 200 days old. Goodger et al. [47] also reported that foliar phenolic compounds were more abundant in young seedlings of *E. froggattii* compared to adult trees. Neither of these studies tagged individual leaves to record the specific age of the foliage used for their analyses.

### 3.4. Synthesis and Application

Photodamage is a potential stress that arises when the quantum of incoming solar radiation greatly exceeds the energy able to be utilised by a leaf’s photosynthetic apparatus. Unused photons generate reactive oxygen species (ROS; [19]), which cause oxidative damage to the leaf’s enzymes, lipids, and pigments, leading to tissue death [17]. ROS are highly reactive forms of oxygen that possess at least one unpaired electron in their orbitals. Under steady-state conditions, ROS are scavenged by various antioxidative defence mechanisms (enzymatic and non-enzymatic). Adaptation to photodamage partly explains why expanding (young) eucalypt leaves are often red in colour under natural light. Our results show that phenolic compounds are abundant in young (three-month-old) *E. camaldulensis* leaves. As oxygen-rich compounds, tannins and flavonols can act as non-enzymatic antioxidants [19,48], while hydrolysable tannins can facilitate photoprotection by preventing anthocyanin degradation [49]. Antioxidants accept electrons from ROS and thereby neutralise them. Phenolic compounds are synthesised by either the combination of the acetate–malonate and shikimate pathways (flavonoids and condensed tannins), or by the shikimate pathway only (hydrolysable tannins such as ellagitannins and gallotannins). Typically, condensed tannins are more abundant in old leaves, while the structurally simpler hydrolysable tannins are more abundant in young leaves [13,50].

We are unable to state whether our treatments, in particular the high-UV_A_ treatment, induced ‘stress’ in our saplings because we took no physiological measurements of foliar photosynthetic efficiency, e.g., as potentially assessed using chlorophyll fluorescence [51]. However, we can state that a number of hydrolysable tannins were more abundant in the leaves grown under our high-UV_A,_ treatment which could represent an adaptive response to this potential stress. It is possible that the severity of the photodamage we induced was relatively mild because the saplings were grown under mild ambient temperatures and were not nutrient-deficient. As previous studies have shown, photoinhibition in eucalypts occurs when leaves are exposed to high levels of light during cold periods and/or when plants are nitrogen-deficient. In *E. moluccana* and *E. fasciculosa*, the symptoms of the photodamage of old (fully expanded) leaves were produced when they leaves experienced these abiotic conditions and their photosynthetic capacity was reduced by the feeding activities of senescence, inducing psyllids belonging to the genus *Cardiaspina* [52,53]. Future studies should aim to include low temperatures and nitrogen limitation as experimental factors to gain a fuller understanding of the effects of elevated UV_A_ on the expression of phenolic compounds by river red gum. An even more complete picture could be provided by also measuring leaf colour, chlorophyll concentrations, and photosynthetic efficiency. Such information is needed because river red gum is widely used in revegetation schemes around Australia and as interventions to increase establishment success (e.g., condition of seedlings, time of planting, type of shelter) can reduce the need for replanting.

## 4. Materials and Methods

### 4.1. Study Species and Growing Conditions

River red gum (*Eucalyptus camaldulensis*) is one of Australia’s most widely distributed species of tree; seven subspecies (abbreviated to ssp.) with distinct distributions are recognised [54]. We purchased five seedlots of the most well-known and widely (globally) planted subspecies, namely, *Eucalyptus camaldulensis* ssp. *camaldulensis*, for our study from the Australian Tree Seed Centre (ATSC), Canberra. These seedlots encompass the entire five-degree latitudinal range of the subspecies, which is characterised by a modest UV_A_ gradient (Figure 6). For example, the parent trees from which these seedlots were harvested span locations varying in annual rainfall from 200 to 600 mm per annum (Table 3). We use the term ‘provenance’ in recognition that the seedlots are from different geographic locations and, hence, that each population has experienced unique conditions of solar radiation, temperature, and rainfall. Such environmental influences have been shown to be associated with the morphological divergence of the leaves of *E. camaldulensis* from disparate geographic locations [55], as well as high genetic differentiation among populations [56]. The terms provenance, genotype, and family are used interchangeably in similar studies of eucalypt PSMs [5,31,57,58,59,60].

Seeds were germinated using Scotts Osmocote Seed and Cutting Premium Potting Mix. One month after germinating, seedlings were transferred to 12 cm pots and maintained in a temperature-controlled glasshouse in the Agriculture Reserve, La Trobe University. The saplings were grown in a native plant potting mix and given 200 mL of water every other day. Branches were bent horizontally and repositioned using bonsai wire to maintain the constancy of light exposure. Frames supporting LED lights were adjusted to keep a constant distance of 50 cm from the tops of the saplings. The temperature was maintained at approximately 21 °C with an 18 h photoperiod. The saplings were rotated two to three times per week so that all plants were equally illuminated.

### 4.2. Experimental Treatments, Leaf Tagging and Harvesting

This study was conducted under glasshouse and ambient (natural light) conditions. Potted saplings were exposed to four treatments, namely, (1) glasshouse with high UV_A,_ (2) glasshouse with low UV_A_, (3) glasshouse with no UV_A_ [control], and (4) ambient light [natural light outside the glasshouse]. The latter two treatments represent a procedural control and a natural reference, respectively. There were 6 saplings (replicates) per provenance for a total of 30 saplings per treatment. There were 120 saplings in total across the four treatments. To block solar, UV transparent UV-absorbing plastic film (SUN 5 Pro) was purchased from Folien-Vertriebs GmbH, Dernbach, Germany, to cover the top of the three 1 m^3^ experimental chambers inside the glasshouse (Figure 7). The spectral properties of the film for UV_A_ and UV_B_ transmittance were measured by an ALMEMO 2390-5 data logger, equipped with UV_A_ and UV_B_ sensors (Ahlborn Mess und Regelungstechnik GmbH, Holzkirchen, Germany). The vertical walls of the experimental chambers were made of black plastic to prevent the sideways illumination of saplings. LED lights were fitted with adjustable chains to maintain a constant height above the plants.

To investigate the effect of leaf age on phenolic composition, branches and petioles were tagged using coloured cable ties to identify individual leaves according to their date of appearance. Tagged leaves from a single branch were harvested after (1) 3 months, (2) 6 months, and (3) 12 months of UV_A_ treatment. Harvested leaves were transferred to labelled paper envelopes and freeze-dried immediately using a Christ Alpha 1–4 LSC freeze drier. They were then stored in the dark prior to analysis.

### 4.3. LED Lighting

We purchased LED lights emitting UV_A_ at the maximum wavelength of 370 nm and LED lights emitting photosynthetically active radiation (PAR) at the wavelength of 400 nm from Shenzhen Vanq Technology Co., Ltd., Shenzhen, China. Two types of UV_A_ LED lights were purchased, namely, (1) one with radiation of 1800 kJ/m^2^ UV_A_ per day, simulating mid-summer UV_A_ [high UV_A_], and (2) one with a radiation of 800 kJ/m^2^ UV_A_ per day simulating, winter annual UV_A_ [low UV_A_]. Light intensity was chosen based on average annual mid-summer and winter radiation data for Yallambie, Victoria (37°726′ S, 145°104′ E; data provided by the Australian Radiation Protection and Nuclear Safety Agency (ARPANSA)). The spectral irradiance and flux of the LED lights were measured at ARPANSA and determined to be suitable for simulating mid-summer and winter UV_A_ when distanced 50 cm from the plants. We assume that high a high level of UV_A_ is create the most stressful of the two experimental light treatments. Two chambers in the glasshouse were fitted with high- and low-UV_A_-LED lights in combination with PAR-LED lights. The control chamber was only fitted with PAR-LED lights. The same types of LED lights were used throughout the study. 

### 4.4. Phenolics Extraction

Freeze-dried leaves were finely ground to ≤0.25 mm using a ball mill (Retsch MM400, Germany) at 30 Hz. Then, 20 mg of finely ground leaf powder was weighed in duplicate into 2 mL microtubes and 1 mL of 80:20 MeOH:H_2_O (*v*/*v*) was added to each sample. The sample was subsequently mixed with a vortex mixer for 2 minutes and sonicated for 10 min (Unisonics, Australia). Samples were then centrifuged for 10 min at 15,000 rpm at room temperature (Eppendorf 5415D bench centrifuge, Hamburg, Germany). The extract was transferred to a clean, labelled microtube. Then, 1 mL of extract was transferred to HPLC vials for liquid chromatography–mass spectrometry (LC-MS) analysis and 5 μL of each extract was used to prepare a pooled sample, which served as a quality control (QC) sample to be analysed at the same time.

### 4.5. Chromatographic and Mass Spectrometric Analysis

Phenolic compounds were identified using ultra-high-performance liquid chromatography (UPLC, UHPLC+ focused, Thermo Scientific™, Waltham, MA, USA) combined with a Q Exactive Hybrid Quadrupole-Orbitrap Mass Spectrometer (Thermo Scientific™, Waltham, MA, USA) and a diode array detector. Data were collected in negative ion mode, scanning a mass range m/z of 100–1500. The negative ion mode was preferred over the positive mode for phenolic compound analysis of all the subgroups because of its sensitivity, clearer fragmentation patterns, and less extensive fragmentation. Nitrogen was used as the sheath, auxiliary, and sweep gas and the spray voltage was set at 3600 V. The capillary temperature was set to 300 °C, with the S-lens RF level set at 64 and an auxiliary gas heater temperature of 310 °C. A HPLC column Hypersil GOLD C18 (175 Å, 3 μm, 2.1 × 150 mm, Thermo Scientific, USA) was used with the column compartment temperature set to 30 °C, and a flow rate was maintained at 0.3 mL/min throughout data acquisition. The mobile phase consisted of (A) 0.1% formic acid and (B) acetonitrile. A linear gradient was used, beginning with 2% of B and reaching 100% of B at 15 min. It was then kept steady at 100% of B until 18 min, and then returned to an initial condition, where it was held for 2 min. The MS/MS analyses were carried out by automatic fragmentation, where the three most intense mass peaks were fragmented. Mass spectrometric (MS) analysis, including the prediction of the chemical formula and exact mass calculation, was performed using Thermo Xcalibur Qual Browser software version 3.0.63 (Thermo Scientific, USA). The samples were use in the mass spectrometry randomly (order organised by using the ‘RAND’ function in Microsoft Excel), with blank and QC samples injected every tenth sample. 

### 4.6. Data Processing

The raw files from Xcalibur were imported to Genedata Expressionist Refiner MS version 12.0, Basel, Switzerland (https://www.genedata.com/). Noise was removed from the data using filters toolsRT structure removal, chemical noise subtraction, peak detection, isotope clustering, adduct detection, singleton filtering, and signal clustering. Further, QC was used for sample normalisation to minimise and correct for batch variation. The cluster-generated data were exported from the Genedata matrix, visualised, and analysed in Genedata Analyst™ 12.0.6 software (Genedata AG, Basel, Switzerland). Genedata Analyst was used for the further integration and interpretation of results.

### 4.7. Data Analysis and Compound Identification

Principal component analysis (PCA) was used to visualise metabolite variation between groups and treatments. PCA is a linear dimensionality reduction methodology that simplifies complexity in high-dimensional data while retaining inherent trends and patterns. The lower dimensions onto which the original data are projected are known as principal components. The first principal component (PC1) is identified as that which minimises the total distance between the data and their projection onto the principal component. By minimising this distance, the variance in the projected points is maximised. The process is repeated, such that the projection of the second PC is uncorrelated with that of the first. We performed a two-group test, using a Student’s *t*-test, in Genedata to compare features between two treatments. *K*-group analysis using ANOVA was used to compare more than two treatments. Highly significant values were ranked based on *p*-values after Bonferroni correction (at *p* = 0.05). For each Bonferroni correction, the critical *p*-value was divided by the number of features, and only the most statistically significant features were selected from the volcano plot produced by Genedata Analyst. These features were tentatively identified using online mass databases, comparison with standards, and MS/MS data. A more complete description of the process of compound identification and a full listing is provided in Chapter II of the work of Khanal [61].

## Figures and Tables

**Figure 1 plants-14-00493-f001:**
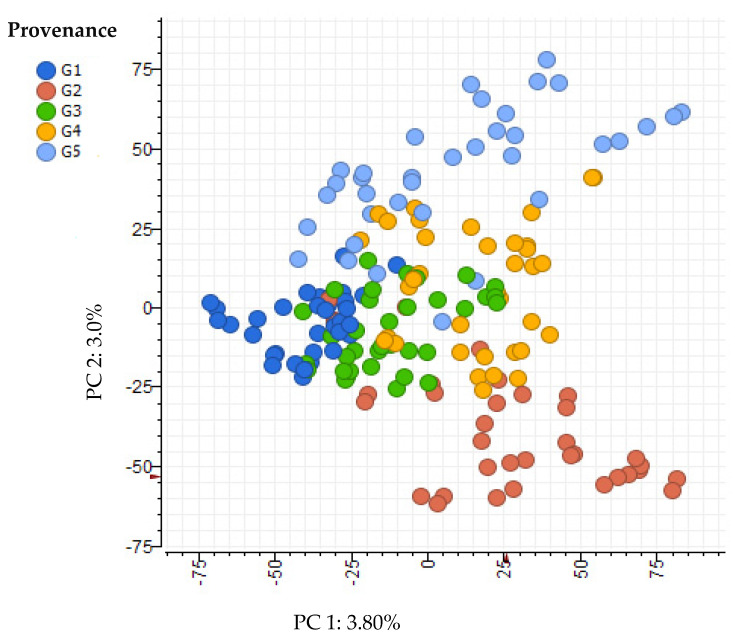
**The** PCA ordination of the phenolic metabolomes of individual saplings, grouped according to provenance. The x-axis plots the value of principal component 1 and y-axis plots the value of principal component 2 for each sapling and provenance.

**Figure 2 plants-14-00493-f002:**
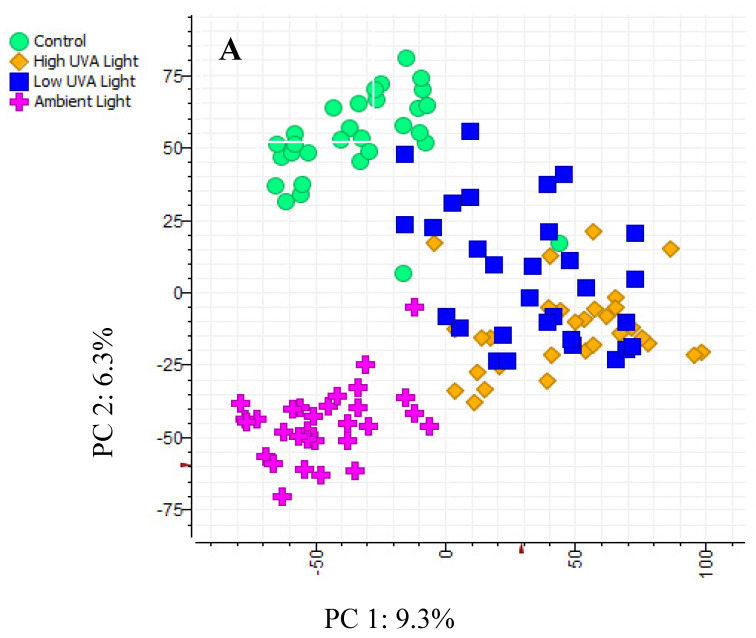
PCA ordinations of the phenolic metabolomes of individual saplings, grouped according to (**A**) high-UV_A_, low-UV_A_, control [glasshouse], and ambient light treatments and (**B**) high-UV_A_ and low-UV_A_ light treatments only. X-axes plot the values of principal component 1 and y-axes plot the values of principal component 2 for each sapling and grouping.

**Figure 3 plants-14-00493-f003:**
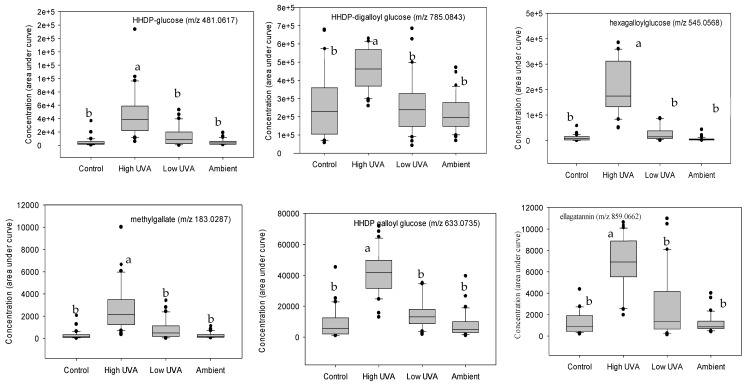
Boxplots showing concentrations of statistically significant phenolic compounds, grouped according to light treatment. X-axes identify the treatment to which the values relate, and the y-axes give the concentrations (as the area under the curve) of the respective compound. In each boxplot, the horizontal line indicates the median, the shaded area indicates the upper and lower quartiles, the whiskers indicate the minimum and maximum values within 1.5× the interquartile range, and black dots are outliers. Treatments with the same superscripted letter are not significantly different (Tukey multiple comparisons of means, *p* < 0.05).

**Figure 4 plants-14-00493-f004:**
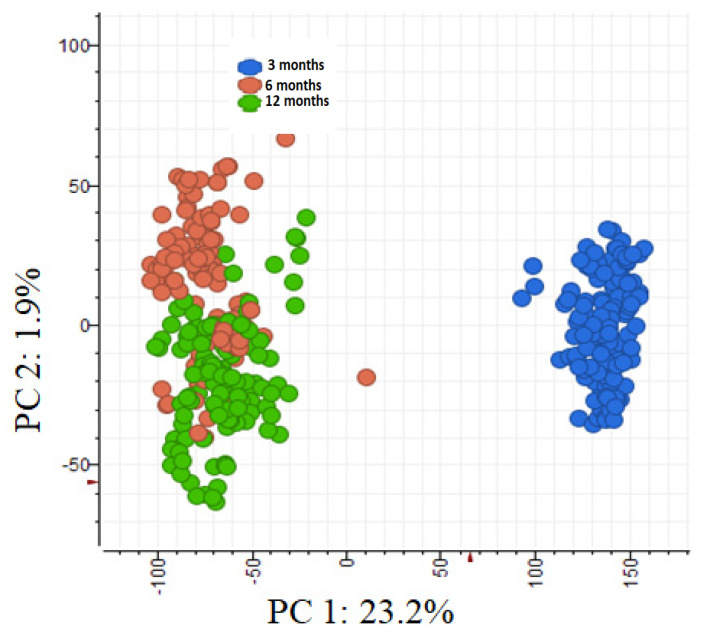
PCA ordination of the phenolic metabolomes of individual saplings grouped according to leaf age (3, 6 and 12 months old). The x-axis plots the value of principal component 1 and the y-axis plots the value of principal component 2 for each sapling and leaf.

**Figure 5 plants-14-00493-f005:**
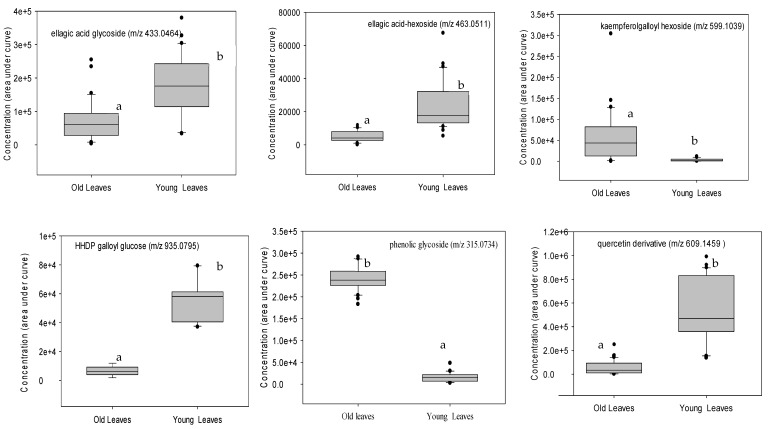
Boxplots showing concentrations of statistically significant phenolic compounds grouped according to leaf age. X-axes identify the age of the leaf to which the values relate and the y-axes give the concentrations (as the area under curve) of the respective compound. See Figure 3 for an explanation of features.

**Figure 6 plants-14-00493-f006:**
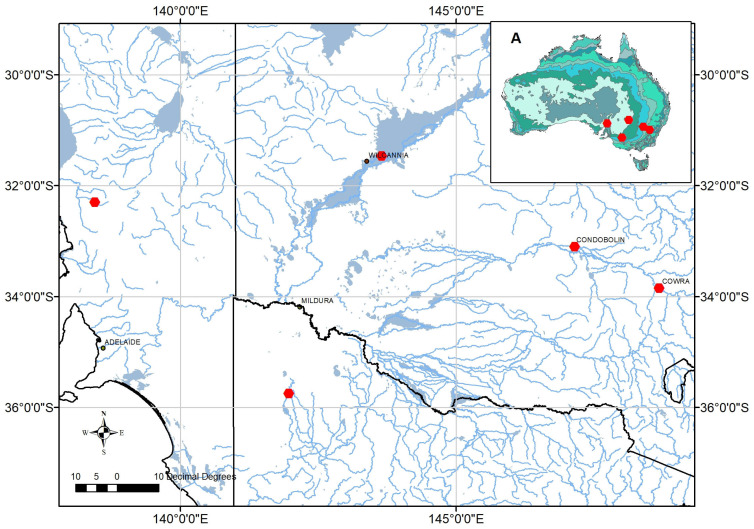
The locations (red dots) referred to in this study. (**A**) An inset map of Australia showing the average annual rainfall contours based on data for the period from 1961 to 1990. (**B**) The locations of parent trees of ATSC seedlots by State (one in Victoria [bottom of map], three in New South Wales [right and centre of map] and one in South Australia [left of map]) relative to proximity to river systems (shown in blue).

**Figure 7 plants-14-00493-f007:**
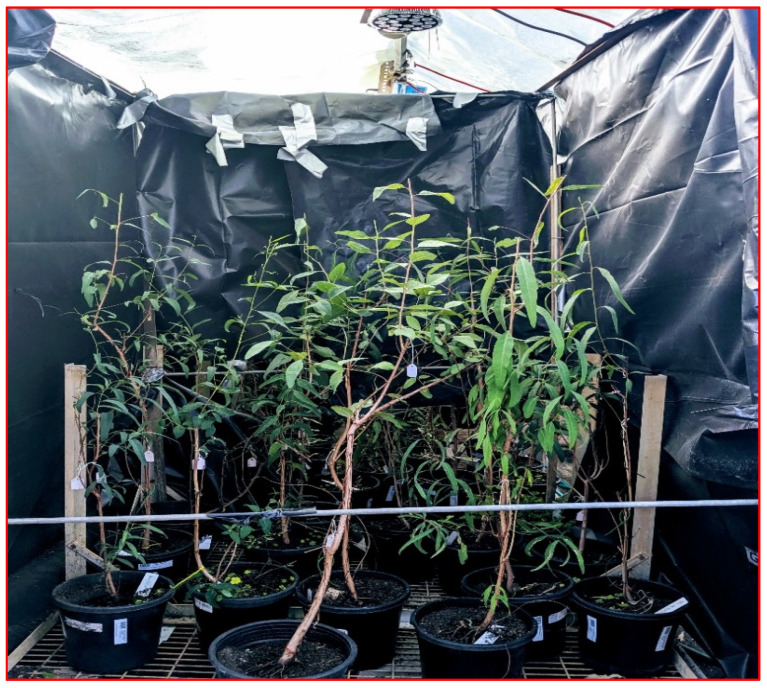
Glasshouse chamber showing *Eucalyptus camaldulensis* saplings under experimental UV_A_ treatment. The top of the chamber was screened from ambient light using UV-absorbing plastic film and black plastic surrounded the four sides to ensure saplings were only exposed to radiation from the LED lights.

**Table 1 plants-14-00493-t001:** Characteristics and tentative identifications of phenolic compounds expressed in statistically significantly (*p* ≤ 0.001) higher concentrations in leaves exposed to high levels of UV_A_.

m/z [M-H]^−^	Retention Time (min)	Fragments	Tentative Identification
1091.1214	5.85	939.1103, 769.0907, 617.0801, 465.0781, 169.0132	hexagalloylglucose I
1091.1214	5.94	939.1103, 769.0907, 617.0801, 465.0781, 169.0132	hexagalloylglucose II
481.0617	2.06	421.0407, 300.9986, 275.0194	HHDP-glucose
1417.1458	3.82	765.0638, 633.0735, 300.9983,275.0201	ellagitannin dimer
545.0568	5.91	939.1103, 769.0907, 617.0801, 465.0781, 169.0132	hexagalloylglucose
785.0843	5.42	633.0736, 300.9991, 275.0198, 249.0399, 169.0133	HHDP-digalloylglucose
104.0131	5.47	~	unknown
331.0669	5.8	271.0459, 169.0132, 151.0012, 125.0234	galloylglucose
783.0686	3.37	481.0619, 300.9984, 275.0196	di-HHDP-glucose
787.0984	4.63	635.0892, 465.0701, 300.9982, 249.0401, 169.0131	tetragalloylglucose
183.0287	5.26	168.0053, 124.0153	methylgallate
433.1725	5.21	~	unknown
785.0843	4.04	633.0736, 300.9991, 275.0198, 249.0399, 169.0133	HHDP-digalloylglucose
633.0735	5.45	463.0533, 300.9985, 275.0201, 169.0135	HHDP-galloylglucose
185.0377	4.15	~	unknown
124.0147	4.17	~	unknown
859.0662	4.33	300.9985, 275.0201, 169.0135	ellagitannin
860.0802	4.34	300.9985, 275.0201, 169.0135	ellagitannin
787.0984	4.41	635.0892, 465.0701, 300.9982, 249.0401, 169.0131	tetragalloylglucose
483.0775	4.13	331.0666, 271.0457, 169.0132, 125.0232	digalloylglucose

**Table 2 plants-14-00493-t002:** Characteristics and tentative identifications of phenolic compounds expressed in statistically significantly (*p* ≤ 0.001) higher concentrations in either young or old leaves. ↑ indicates the increased relative concentration.

m/z [M-H]^−^	Retention Time (min)	Tentative Identification	Young Leaves	Old Leaves
383.1865	15.33	unknown		↑
433.0464	5.26	ellagic acid glycoside	↑	
271.082	3.65	glycosylated hydroquinone		↑
463.0511	4.72	ellagic acid hexoside	↑	
433.0462	5.27	ellagic acid glycoside	↑	
599.1039	5.49	kaempferol derivative		↑
315.0734	3.98	phenolic glycoside		↑
197.0812	3.65	unknown	↑	
499.1602	14.21	unknown		↑
783.0693	3.30	pedunculagin		↑
461.0720	5.85	flavonoid glycoside	↑	
935.0795	5.51	di-HHDP galloylglucose	↑	
789.4441	16.02	unknown		↑
471.2746	13.32	unknown	↑	
399.1814	15.01	unknown		↑
609.1459	5.10	quercetin derivative	↑	

**Table 3 plants-14-00493-t003:** Provenances of *Eucalyptus camaldulensis* ssp. *camaldulensis* used in this study.

ATSC Seedlot Number	Code	GPS Coordinates	Location (State)	Average Annual Rainfall (mm)
20440	G_1_	−31.46 S, 143.65 E	30 km ENE of Wilcannia (New South Wales)	264
20561	G_2_	−35.75 S, 141.96 E	Lake Albacutya (Victoria)	363
20437	G_3_	−32.30 S, 138.45 E	Boolcunda Creek (South Australia)	306
20430	G_4_	−33.85 S, 148.70 E	Cowra (New South Wales)	598
20429	G_5_	−33.10 S, 147.15 E	Condobolin (New South Wales)	421

## Data Availability

Data are contained within the article.

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
