# Peer review of "Ultraviolet-A Radiation (UVA) as a Stress and the Influence of Provenance and Leaf Age on the Expression of Phenolic Compounds by Eucalyptus camaldulensis ssp. camaldulensis"

_plants, 2025, doi:10.3390/plants14030493_

Round 1

Reviewer 1 Report

Comments and Suggestions for Authors

The manuscript entitled “UVA as a Stress and the Influences of Provenance and Leaf Age on the Expression of Phenolic Compounds by Eucalyptus camaldulensis ssp. Camaldulensis” depicted the effects of different intensity of UVA, Provenances, and leaf age process on phenolic compounds of river red gum from different provenance. The work is interesting and the experiment is well-designed. My suggestion is accepted. Here are some suggestions which might be able to help with the improvement of the manuscript.

The title should be revised to fit the content of the manuscript.

The full name of UV, UVA, and UVB must be mentioned in the title, abstract, and text.

Line 14: What do you mean by unbiased metabolomics? I supposed it should be untargeted or targeted metabolomics.

Some precise values from the results should be added to the abstract.

The keywords seem to be too much.

Figure 1: The PCA only explained less than 10% of the variation. Is that really need to be represented in the results? Similar problems for other PCA.

3.4: What do you mean by summary? I suppose it is like conclusion. However, conclusion should be expressed as an individual part not a part of the results. Furthermore, the length seems to be too long and too much paragraph.

4.4 and 4.5: The determinations lacks of references.

Author Response

Reviewer 1

Comments 1: The title should be revised to fit the content of the manuscript.

Response 1: We do not agree with this suggestion. Neither of the other two Reviewers requested such a modification. No change.

Comments 2: The full name of UV, UVA, and UVB must be mentioned in the title, abstract, and text.

Response 2: “UV” is the commonly understood and used abbreviation for ultraviolet radiation (see https://en.wikipedia.org/wiki/Ultraviolet). Nevertheless, we have included the full term (along with its respective abbreviation) when first used in the title and abstract but not in the text so as to avoid repetition.

Comments 3: Line 14: What do you mean by unbiased metabolomics? I supposed it should be untargeted or targeted metabolomics.

Response 3: Unbiased is an alternate term for untargeted and was understood as such by Reviewer 2. We have included the word “untargeted” in the abstract.

Comments 4: Some precise values from the results should be added to the abstract.

Response 4: We consider that inclusion of means in the abstract would reduce the readability of the text and prevent readers from easily obtaining an understanding of plant responses. No change.

Comments 5: The keywords seem to be too much.

Response 5: We have adhered to the recommended number of keywords. We do not agree with this suggestion. No change.

Comments 6: Figure 1: The PCA only explained less than 10% of the variation. Is that really need to be represented in the results? Similar problems for other PCA.

Response 6: This finding is not a “problem”. The fact that provenance and treatment explained less variation than leaf age is a key finding of the study. Hence, all ordinations should be shown to emphasise these relationships. It should also be noted that, although relatively little variation is explained by the ordination in Fig. 1, it is possible to see clear groupings of the provenances. The same applies to the ordinations provided for Fig. 2. No change.

Comments 7: 3.4: What do you mean by summary? I suppose it is like conclusion. However, conclusion should be expressed as an individual part not a part of the results. Furthermore, the length seems to be too long and too much paragraph.

Response 7: If the reviewer does not appreciate the role of this subtitle at the end of the Results section, perhaps they will appreciate “Synthesis and application” instead. The subtitle used has been changed in the text. We do not agree that section 3.4 is too long given the role of this section of the Results. No change to length.

Reviewer 2 Report

Comments and Suggestions for Authors

This paper fills a gap in the existing literature regarding the responses of eucalyptus trees to environmental stressors, particularly their tolerance to the intense Australian sun. By acknowledging this knowledge gap, the author aims to contribute to the understanding of the ecological and economic significance of these trees, which are a vital component of the Australian ecosystem.

The author describes a study on the responses of Eucalyptus camaldulensis to UVA radiation using a cutting-edge, unbiased metabolomics approach. This innovative method allows the author to provide new insights into the phytochemistry of eucalypts and to explore how variations in their foliar phenolics may relate to their tolerance of solar radiation. The author's aim is to shed light on the mechanisms underlying the tolerance of these trees to high levels of solar radiation.

In summary, the author seeks to enhance knowledge about eucalypt responses to environmental stress while employing novel methodologies to achieve this goal.

However, several issues need to be addressed in the paper.

At the beginning of the paper, the author should introduce and explain the abbreviations used throughout the text, rather than expecting the reader to be familiar with them.

Clarify the meaning of components PC 1 and PC 2, and define what PCA (principal component analysis) is, especially since it is mentioned in the Materials and Methods section, which is typically located at the end of the paper.

Consider reorganizing the paper to include a clear description of the methods used in the study at the beginning, rather than at the end.

The figure legends should be rewritten to provide a clear and concise explanation of the data presented in the figures, including the meaning of the X and Y axes.

The figure numbers should be corrected, as there are multiple figures with the same number. For example, the second figure with the number 4 should be renumbered as figure 5.

Figure 3 and the second figure with the number 4 should be resized to make it easier to read and interpret. Consider using a scale dependency to show the relationship between phenolic compounds and age (time).

The tables should be redesigned to present the data clearly, rather than using traditional numerical formats. Consider including graphical representations of the chromatographic and mass spectrometric data.

The field experiments and laboratory conditions should be related to each other. In addition photosynthetic data should be included to support the conclusions drawn in the paper.

Consider revising the Introduction and Discussion sections to remove unnecessary sentences and statements.

Author Response

Reviewer 2

Comments 1: At the beginning of the paper, the author should introduce and explain the abbreviations used throughout the text, rather than expecting the reader to be familiar with them.

Response 1: Abbreviations all now accompanied by full translation (as per suggestion of Reviewer 1).

Comments 2: Clarify the meaning of components PC 1 and PC 2, and define what PCA (principal component analysis) is, especially since it is mentioned in the Materials and Methods section, which is typically located at the end of the paper.

Response 2: Translations of PCA (provided in M&Ms but not Results), PC1 and PC2 provided in section 2.1 of Results. Description of PCA now provided in M&Ms. We assumed that all researchers understood this basic statistical technique hence the reason for our brevity.

Comments 3: Consider reorganizing the paper to include a clear description of the methods used in the study at the beginning, rather than at the end.

Response 3: We followed the guidelines to authors provided by Plants in the layout of our paper. No change.

Comments 4: The figure legends should be rewritten to provide a clear and concise explanation of the data presented in the figures, including the meaning of the X and Y axes.

Response 4: Neither Reviewer 1 nor 3 made this suggestion, i.e. they apparently understood our figure legends. Nevertheless, we have modified the captions of each figure to facilitate interpretation of our figures.

Comments 5: The figure numbers should be corrected, as there are multiple figures with the same number. For example, the second figure with the number 4 should be renumbered as figure 5.

Response 5: Corrected.

Comments 6: Figure 3 and the second figure with the number 4 should be resized to make it easier to read and interpret. Consider using a scale dependency to show the relationship between phenolic compounds and age (time).

Response 6: The scales of the y-axes used in Figures 3 and 4 (re-numbered as Figure 5) are determined by the concentrations of the respective compounds. Some compounds were more concentrated than others and, for some compounds, there was greater variation in individual values which is shown by the inclusion of outliers. Hence, we cannot standardise the y-axes without compromising the illustration of values for some compounds. No change. We do not see the point of providing a relative scale on the x-axis (presumably) of Fig 5 between two categories, i.e. between “young leaves” (3-months old) and “old leaves” (combination of 6- and 12-month old leaves). No change.

Comments 7: The tables should be redesigned to present the data clearly, rather than using traditional numerical formats. Consider including graphical representations of the chromatographic and mass spectrometric data.

Response 7: We disagree that the numerical data presented in our Tables are unclear. Indeed, neither Reviewer 1 nor 3 made this suggestion. No change.

Comments 8: The field experiments and laboratory conditions should be related to each other. In addition photosynthetic data should be included to support the conclusions drawn in the paper.

Response 8: There was no “field experiment”. As described in the M&Ms section, the saplings grown under ambient (natural) light were our “natural reference” to ‘scale’ changes in the phenolic metabolomes induced by our experimental, glasshouse treatments, including of our procedural control saplings. As we state in the Discussion, we did not take any measurements of the photosynthetic efficiency of the saplings. This was an oversight on our part. No change.

Comments 9: Consider revising the Introduction and Discussion sections to remove unnecessary sentences and statements.

Response 9: We do not feel that either our Introduction or Discussion is verbose. Indeed, our Introduction is possibly too brief considering the wealth of information on the topic of plant stress responses to UV. We deliberately sought to be as concise as possible throughout the manuscript. Reviewer 3 wrote “the writing and presentation is quite clear”. No change.

Reviewer 3 Report

Comments and Suggestions for Authors

This seems to be a well-designed and conducted experiment, and the writing and presentation is quite clear.  My only suggestion is to provide the photosynthetic photon flux density of the PAR-LED lamps.  Also, are they really monochromatic at 400nm?  Why not use a broader spectrum PAR-LED lamp system?  These are minor points.

Author Response

Reviewer 3

Comments 1: My only suggestion is to provide the photosynthetic photon flux density of the PAR-LED lamps.  Also, are they really monochromatic at 400nm?  Why not use a broader spectrum PAR-LED lamp system?  These are minor points.

Response 1: We were not able to measure the photon flux density of the PAR-LED lamps. We report in our paper the specifications provided for these LEDs by Shenzhen Vanq Technology Co., Ltd., Guangdong Sheng, China. We had limited grant funds for the experiment and could not consider purchasing other PAR-LED lamps. No change. We have improved the English expression starting at line 322.

Round 2

Reviewer 2 Report

Comments and Suggestions for Authors

Accept